

# Mountain colonisation, miniaturisation and ecological evolution in a radiation of direct-developing New Guinea Frogs (*Choerophryne*, Microhylidae)

Paul M. Oliver[1], Amy Iannella[2], Stephen J. Richards[3] and Michael S.Y. Lee[3,4]

[1] Division of Ecology and Evolution, Research School of Biology & Centre for Biodiversity Analysis, Australian National University, Canberra, Australian Capital Territory, Australia

[2] School of Biological Sciences, The University of Adelaide, Adelaide, South Australia, Australia

[3] South Australian Museum, Adelaide, South Australia, Australia

[4] School of Biological Sciences, Flinders University, Adelaide, South Australia, Australia

Corresponding author
Paul M. Oliver,
paul.oliver@anu.edu.au

## ABSTRACT

**Aims**. Mountain ranges in the tropics are characterised by high levels of localised endemism, often-aberrant evolutionary trajectories, and some of the world's most diverse regional biotas. Here we investigate the evolution of montane endemism, ecology and body size in a clade of direct-developing frogs (*Choerophryne*, Microhylidae) from New Guinea.

**Methods**. Phylogenetic relationships were estimated from a mitochondrial molecular dataset using Bayesian and maximum likelihood approaches. Ancestral state reconstruction was used to infer the evolution of elevational distribution, ecology (indexed by male calling height), and body size, and phylogenetically corrected regression was employed to examine the relationships between these three traits.

**Results**. We obtained strong support for a monophyletic lineage comprising the majority of taxa sampled. Within this clade we identified one subclade that appears to have diversified primarily in montane habitats of the Central Cordillera (>1,000 m a.s.l.), with subsequent dispersal to isolated North Papuan Mountains. A second subclade (characterised by moderately to very elongated snouts) appears to have diversified primarily in hill forests (<1,000 m a.s.l.), with inferred independent upwards colonisations of isolated montane habitats, especially in isolated North Papuan Mountains. We found no clear relationship between extremely small body size (adult SVL less than 15 mm) and elevation, but a stronger relationship with ecology—smaller species tend to be more terrestrial.

**Conclusions**. Orogeny and climatic oscillations have interacted to generate high montane biodiversity in New Guinea via both localised diversification within montane habitats (centric endemism) and periodic dispersal across lowland regions (eccentric endemism). The correlation between extreme miniaturisation and terrestrial habits reflects a general trend in frogs, suggesting that ecological or physiological constraints limit niche usage by miniaturised frogs, even in extremely wet environments such as tropical mountains.

## INTRODUCTION

Tropical mountains contain some of the most diverse regional biotas in the world, with high levels of localised endemism and often fine elevational turnover in biodiversity (*Mayr & Diamond, 1976*; *Fjeldså, Bowie & Rahbek, 2012*; *Merckx et al., 2015*; *Rosauer & Jetz, 2015*). The origins of, and processes shaping, this exceptional diversity are of great scientific interest, both for improved understanding of the drivers of biological diversity (*Janzen, 1967*; *Hutter, Guayasamin & Wiens, 2013*; *Graham et al., 2014*), and for understanding how these highly diverse biotas will be affected by anthropogenic climatic change (*Williams, Bolitho & Fox, 2003*; *La Sorte & Jetz, 2010*; *Freeman & Class Freeman, 2014*).

Two broad paradigms to explain high diversity in tropical mountains have been advanced (*Fjeldså, Bowie & Rahbek, 2012*), and both received support from different studies: (a) mountain uplift and climatic change have driven local speciation (the 'cradle' hypothesis; *Weir, 2006*; *Price et al., 2014*), or (b) mountains have provided refugia, often for specialised taxa that would have otherwise died out due to competition or climatic change (the 'museum' hypothesis; *Hutter, Guayasamin & Wiens, 2013*). In a recent study focused on understanding the biogeographic origins of montane endemics *Merckx et al. (2015)*, suggested they could also be broadly dichotomised into centric endemics (derived from upslope colonisation of lowland taxa) and eccentric endemics (derived via long distance colonisation of cool adapted taxa).

The large tropical island of New Guinea has arguably the 'most complex orogeny in the world' (*Baldwin, Fitzgerald & Webb, 2012*). The collision of the leading edge of the northwards-moving Australian plate with the westwards-moving southern edge of the Pacific Plate has uplifted a high Central Cordillera (>4,000 m a.s.l.) extending nearly the length of the island (*Baldwin, Fitzgerald & Webb, 2012*) (Fig. 1A). These ranges may date from the late Miocene, and high elevation habitats are even younger (*Hall, 2002*; *Van Ufford & Cloos, 2005*; *Baldwin, Fitzgerald & Webb, 2012*). Beginning in the Miocene, and continuing with the ongoing rapid uplift of the Huon and Finnisterre Ranges (Fig. 1), additional smaller and more isolated montane regions scattered along northern New Guinea are the uplifted remnants of island arcs that have accreted onto the northern edge of the Australian plate (*Hall, 2002*; *Polhemus, 2007*).

The biota of New Guinea has been profoundly shaped by this complex orogeny. The uplift of the Central Cordillera has largely isolated the biotas of lowland regions to the north and south of New Guinea (*Rawlings & Donnellan, 2003*; *Unmack, Allen & Johnson, 2013*; *Georges et al., 2014*). It has also been suggested that emerging elevation gradients may have increased speciation rates in some New Guinea radiations, inflating regional alpha diversity (*Toussaint et al., 2013*; *Toussaint et al., 2014*), a species pump model similar to the uplift of the northern Andes (*Weir, 2006*; *Santos et al., 2009*). In contrast the endemic montane fauna of the smaller, younger and more isolated mountains of northern New Guinea is particularly poorly known, and there have been few phylogenetically-informed assessments of the origins of endemic taxa in these ranges (*Beehler et al., 2012*; *Oliver, Richards & Sistrom, 2012*; *Oliver et al., 2016*).

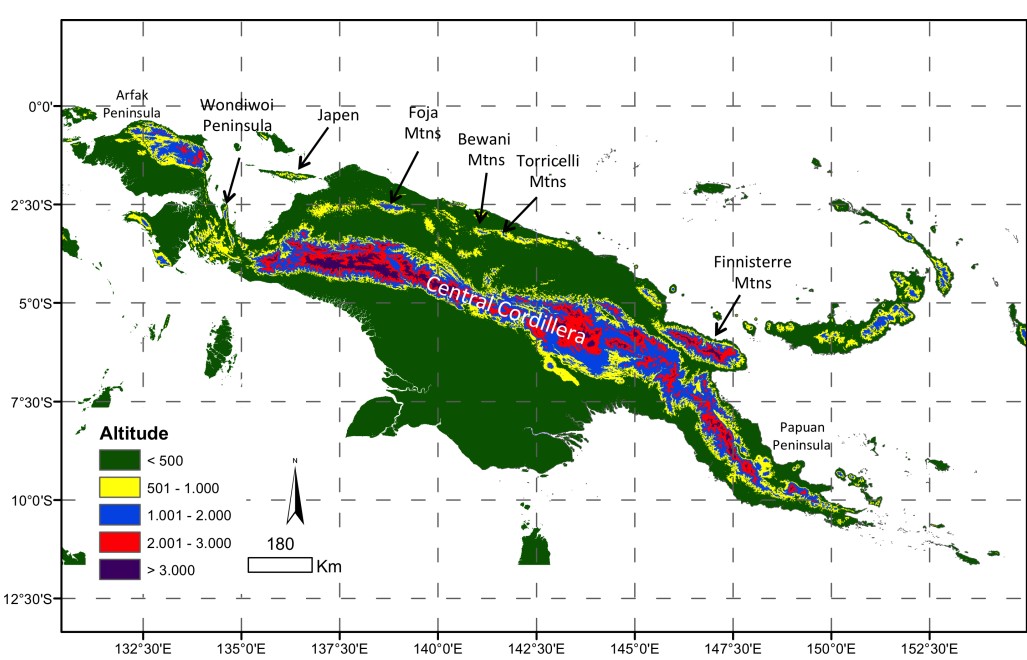

**Figure 1** **Map of New Guinea.** Stratified by major elevation bands and with key areas of montane forest denoted.

The New Guinea frog biota is exceptionally diverse, with >450 recognised species, and many more awaiting description (*Oliver et al., 2013*; *Frost, 2014*)—far more diverse than nearby landmasses such as Borneo or Australia. Such anuran diversity is remarkable for being dominated by just two major radiations, of which the most speciose and ecologically diverse is a clade of nearly 250 recognised species of direct-developing microhylids, the Asterophryinae Günther, 1858 (*Frost et al., 2006*). Their reproductive ecology, wide elevational distribution, high levels of localised endemism, and overall species richness suggest that microhylid frogs may provide an excellent system for understanding how the mountains may have shaped diversification in New Guinea.

*Choerophryne* (including the previously recognised genus *Albericus*: see *Peloso et al., 2015*) is a moderately diverse clade (31 recognised taxa) within the Asterophryinae, comprised of small to miniaturised frogs endemic to New Guinea. This genus occurs from hill to upper montane habitats across much of the Central Cordillera and North Papuan Mountains (although they appear to be largely absent from most of the west and southern lowlands of the island) (*Günther, 2000*; *Richards, Iskandar & Allison, 2000*). Broadly, taxa formerly placed in the genus *Albericus* are mostly climbing frogs with well-developed finger and toe pads, while taxa formerly placed in *Choerophryne* tend to be more terrestrial, although there are exceptions to this general trend (*Kraus & Allison, 2000*; *Richards, Dahl & Hiaso, 2007*; *Günther & Richards, 2011*) (Fig. 2).

*Choerophryne* also includes many miniaturised species, here defined as frogs less than 15 mm long (*Yeh, 2002*) and approaching minimum size limits for tetrapods (*Kraus, 2010a*; *Rittmeyer et al., 2012*). The water-permeable skin of frogs plays a critical role in shaping both local and regional patterns of diversity and habitat use (*Scheffers et al., 2013*),

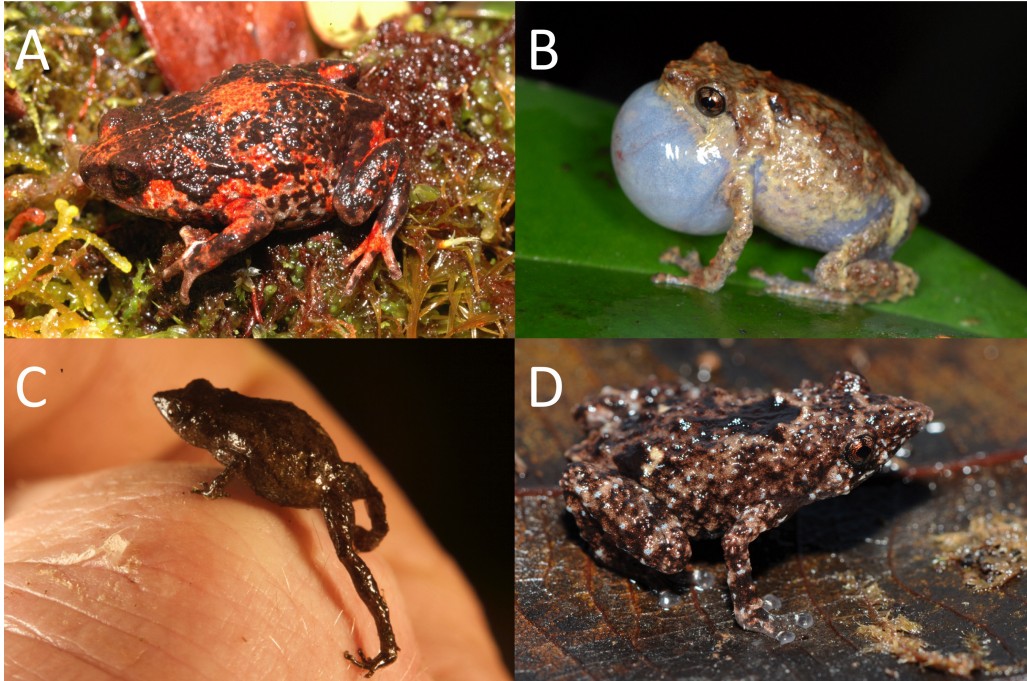

**Figure 2  Representative species of *Choerophryne*.** (A) *Choerophryne alpestris* upper montane moss fields, Central Cordillera, terrestrial; (B) *Choerophryne* spA7 hill forest, southern foothills, scansorial; (C) *Choerophryne* spB1 lower montane forest, Foja Mountains, terrestrial; (D) *Choerophryne proboscidea* hill forest forest, northern lowlands, scansorial. Photographs courtesy S Richards (A, B, D) and T Laman (C).

with smaller species at greater risk of desiccation than larger species (*Tracy, Christian & Richard Tracy, 2010*). It follows therefore, that smaller size in *Choerophryne* species may be correlated with occurrence in more humid environments. We hypothesise that such a trend may manifest as a negative correlation between body size and altitude, due to the existence of reliably moist cloud forest habitats at higher elevations.

Here we present an analysis of the phylogenetic relationships and evolution of key traits within *Choerophryne*. We initially focus on the origins of montane endemism, with a specific prediction being that the older Central Cordillera will be dominated by *in situ* diversification processes (centric endemism) linked to ongoing uplift, while the younger North Papuan mountains may show evidence of colonisation from the older Central Cordillera (eccentric endemism). We also test the prediction that ecological shifts (arboreal to terrestrial), and shifts in body size (towards extreme miniaturisation) may correlate with occurrence in novel habitats and climatic regimes at higher elevations.

## METHODS

### Specimen selection

This study utilised whole specimens and tissue samples deposited in Museum collections (ethics approval was therefore not required). Full details of samples included are given in Tables S1–S2. In our final analysis we included 18 of 31 recognised species, plus 11 candidate taxa. Following *Vieites et al. (2009)* we recognised lineages as distinct OTUs (candidate

species) for downstream analysis if they met any two of the following three criteria: (a) males with distinctive advertisement calls, (b) evidence of morphological differentiation or (c) evidence of genetic differentiation (usually greater than 3% uncorrected pairwise in the 16S rRNA gene (see Table S3 for a summary). Mitochondrial DNA sequences of an additional 11 *Choerophryne* were downloaded from GenBank, along with 14 outgroup sequences from 6 other New Guinean microhylid genera. The taxonomic assignation of *Choerophryne* species is challenging, especially in the absence of calls, so taxonomic designations used in this study should be considered provisional. It is possible further work may demonstrate that some candidate species are conspecific with additional recognised taxa. Full details of samples included are given in Appendix S1.

### DNA extraction, amplification, sequencing and alignment

Whole genome DNA was extracted from frozen or alcohol preserved liver samples using the Gentra Puregene kit protocol (QIAGEN 2011). Sequence data from the 12S and 16S mitochondrial genes was PCR amplified with an annealing temperature of 58 °C using the primers 12SAL and 12SBH (*Palumbi et al., 2002*) and 16SL3 and 16SAH (*Vences et al., 2003*), then purified on MultiScreen PCR$_{384}$ Filter Plates. Sanger sequencing (forward and reverse) of purified PCR product used the BigDye Terminator v3.1 Cycle Sequencing Kit (Applied Biosystems), purified using MultiScreen SEQ$_{384}$ Filter Plates and sent to the Australian Genome Research Facility (AGRF) for capillary separation.

Geneious Pro v5.5.2 (*Kearse et al., 2012*) was used to align forward and reverse sequence traces and reviewed by eye. The consensus sequences along with sequences from GenBank (Appendix S1) were aligned with 8 iterations of the MUSCLE algorithm under default parameter settings (*Edgar, 2004*). Hypervariable regions with poor local alignment were removed using Gblocks v0.91b (*Castresana, 2000*); of the original 1,556 aligned positions, 1,347 were retained in final analyses.

### Phylogenetic analysis

To assess congruence of topology and support values across methods, we estimated phylogenetic relationships using Bayesian and maximum likelihood approaches. Based on the output of the model selection program MrModeltest (*Nylander, 2004*) all analyses were performed using the general time-reversible model, allowing for variation in the rate of evolution among sites and including invariable sites (GTR + I + G). Both genes were treated as a single partition due to the relatively short sequence length and similar features (i.e., mitochondrial rRNA).

The maximum likelihood tree with bootstrap values was produced using RAxML v 8.0.26 (*Stamatakis, 2006*) with bootstrap scores calculated using the rapid bootstrap (-f a) function with 1,000 replicates. The Bayesian consensus tree was generated by Mr Bayes 3.2.2 (*Ronquist et al., 2012*) using an unconstrained branch length prior, 4 chains (incrementally heated at temperature 0.2), each of 5 million generations with a 1 million generation burn-in and sampling every 200 generations.

These topology-only analyses with dense sampling across populations were compared to analyses where we simultaneously estimated phylogeny, divergence dates and trait evolution on species-level trees (see below).

## Trait and biogeographical scoring

We scored each taxon for three traits of interest: (i) adult male body size, (ii) elevation and (iii) maximum calling height of males (as a proxy for arboreality vs terrestriality) (Table S4). These data were scored from genotyped specimens and associated field notes, or extracted from primary literature.

We used a typical measure of size in anurans, the distance from the tip of the snout to the urostyle tip (SUL), which has been previously used in *Choerophryne* (*Günther, 2008*). We used the maximum recorded size for males (sex determined by observations of specimens calling). Although some *Choerophryne* have unusually long snouts, at most these comprised 10% of the total body length.

The maximum elevational range (difference between lower and upper occurrences) obtained for any species was just over 1,000 m, involving three taxa that occur primarily in hill forests, but range into lower montane forests (*Choerophryne gracilirostris, C. rostellifer* and *C.* sp A7). Seven taxa are also only known from single sites. To score elevation as a continuous character (for use in phylogenetic regressions) we used the mid-point of records for each lineage (to the nearest 100 m).

For discrete categorisation of elevation we used the forest classification system presented by *Johns (1982)*: hill forest and lowlands (<1,000 m a.s.l.), lower montane (1,000–2,000 m a.s.l.), mid-montane (2,000–3,000 m a.s.l.) and upper montane (>3,000 m a.s.l.). These bands broadly reflect how reducing mean temperatures with elevation shapes the transition from megathermal to microthermal vegetative communities (*Nix, 1982*). For most taxa, the majority of records were focused in just one of these bands. The small number of taxa whose distributions spanned bands were placed in the band in which the majority of records were concentrated. Finally, *Choerophryne laurini* is known only from typical lower montane forest on mossy ridge tops in the Wondowoi mountains between 800–950 m, although it may also occur in lower montane forest in the Snow Mountains (*Richards et al., 2015*). This species was coded as lower montane for discrete analyses.

To better visualise potential colonisation paths to the isolated North Papuan Mountains, we also devised a further coding system of four states that combined geography and elevation: southern lowland (south of Central Cordillera below 1,000 m a.s.l.), central highlands (Central Cordillera above 1,000 m a.s.l.), northern lowland (north of Central Cordillera below 1,000 m a.s.l.) and northern montane (North Papuan Mountains above 1,000 m a.s.l).

Male *Choerophryne* show extensive variation in their typical calling height, from largely terrestrial (e.g., *Choerophryne alpestris*) to more than 3 metres off the ground (e.g., *Choerophryne pandanicola*) (*Günther & Richards, 2011*). To score calling height as a continuous trait we used the maximum recorded calling height of males, either from the literature or personal observations. We also employed a second scheme for coding calling ecology, by dividing taxa into two broad guilds: (a) *Terrestrial*— species that called predominantly on or very close to the ground on leaf litter or duff (generally less than 50 cm high); and (b) *Scansorial* —species that usually climb into vegetation and call from (generally more exposed) positions up to several metres high. Two taxa (*C. arndtorum* and *C. microps*) for which the majority of calling records are terrestrial but which have

occasionally been recorded calling a metre or more above the ground (*Günther, 2008*), were coded as terrestrial in the discrete character analyses, while the maximum recorded calling height was used in continuous trait based analyses.

## Ancestral state analyses

We used BEAST v 1.8.2. (*Drummond & Rambaut, 2007*) to co-estimate trait evolution (including ancestral states) with phylogeny and divergence dates. These analyses used a reduced dataset comprising a single exemplar of each genetically and/or morphologically divergent lineage identified in earlier phylogenetic analyses (i.e., recognised or candidate species). The original molecular data for each exemplar was also included. To ensure these analyses were focused on a strongly supported and well-sampled monophyletic group, in these trait analyses we excluded two samples from a highly divergent clade that did not strongly associate with other *Choerophryne* in estimated phylogenies (see results). Size was $\log_{10}$ transformed. The two discrete variables (elevation and calling ecology) were coded using the MK + strict clock model, which assumes that transformations between states are reversible and occur at the same rate throughout the tree; more complex models were not feasible due to the relatively small tree and number of transformations. Elevation character states were ordered—e.g., shifts from lower- to upper-montane habitats were constrained to involve moving through intervening mid-montane habitats. Analyses were run for 50 million generations, sampling every 50,000 generations. The first 20% of trees were discarded as burnin and the remaining 800 post-burnin trees from each run were combined to generate the final consensus topology. The final xml file is in Appendix S2. Effective samples sizes (ESS) for all parameters (Tracer v 1.6.0; *Drummond & Rambaut, 2007*) in both individual and combined BEAST analyses were above 200.

BEAST automatically produces an ultrametric tree, however there are no fossil calibrations within *Choerophryne*, and there has been no recent thorough assessment of rates of mitochondrial DNA evolution in frogs. To provide a rough timescale for *Choerophryne,* we used a broad consensus molecular evolutionary rate for mitochondrial genes of between 1 and 2% pairwise divergence per million years, which was incorporated into the prior for the average substitution (clock) rate. Rates of molecular variation vary extensively (*Eo & DeWoody, 2010*), and thus the resultant dates from this analysis are interpreted with caution. Importantly, the ancestral state analyses (above) only require relative rather than absolute branch lengths (e.g., they could still be performed if root age was arbitrarily scaled to 1), so our results are robust to these dating uncertainties.

## Phylogenetic least squares regression

The relationship of (a) body size to calling ecology and/or elevation, and (b) calling ecology to elevation was analysed using BayesTraits v 2.0 (*Pagel & Meade, 2013*) across the concatenated 3,200 post-burnin trees from BEAST. For these analyses all variables were included as $\log_{10}$-transformed continuous states. We only included data for lineages in two well-sampled clades of *Choerophryne* that were strongly supported as sister taxa (see below); other species in the trees were scored as missing data. We also performed regressions on each of these two well-differentiated clades. The Bayesian MCMC implementation of the

continuous module was used to regress (a) body size against ecology and elevation; and (b) ecology against elevation. Eleven million steps were used with the first 1 million discarded for burnin, and 4 runs of BayesTraits were performed and checked for convergence using Tracer v 1.6.0 (*Drummond & Rambaut, 2007*). *Pagel & Meade (2013)* state that the significance of a variable can be assessed either by comparing harmonic means (for analyses with and without the variable), or observing whether the estimated distribution of that variable (e.g., 95% HPD) excludes 0. Due to issues around the use of harmonic means to estimate marginal likelihoods (*Xie et al., 2011*), we adopted the latter approach.

## RESULTS

### Phylogenetic relationships and lineage diversity

Bayesian and maximum likelihood analyses identified three major lineages of *Choerophryne* (Fig. 3, Fig. S1). Clade A comprised the majority of sampled taxa that were formerly placed in the genus *Albericus*, Clade B included all taxa with a moderate to pronounced rostral projection formerly placed in *Choerophryne sensu stricto*. Clade C comprised two scansorial taxa lacking distinctive rostral projections that occur south of the Central Cordillera in hill forest, and on the Finnistere Ranges (north-east New Guinea) in hill to lower montane forest.

A sister taxon relationship between Clades A and B was strongly supported in all analyses (Posterior Probability 1.0, bootstrap support > 90). Clade C was more divergent and there was no evidence that it forms the sister group to Clade A + B (or any other microhylid lineage). All basal relationships between the sampled New Guinea microhylid genera were poorly supported, but these were not the focus of this study.

Within Clade A we identified two strongly supported primary lineages, with the major split being between a clade of two lower montane and hill forest taxa from the south side of the Central Cordillera, and several clusters of species from across the Central Cordillera and North Papuan Mountains, including derived terrestrial taxa from mid to upper montane habitats (*C. alpestris* and *C. brevicrus*).

Within Clade B there were three well supported primary lineages: one comprising three deeply divergent taxa (*C. burtoni* and two unnamed taxa) from hill forest to mid-montane habitats on the Central Cordillera; a further lineage of large-bodied and very long-snouted taxa from hill and lower montane forest in northern New Guinea; and finally a diverse conglomeration including lineages from hill and lower montane forests in northern New Guinea, in addition to one taxon from south of the Central Cordillera (*C. gracilirostris*).

In all three major clades we identified lineages (candidate species) that were deeply divergent from, and could not be confidently assigned to, recognised species. This was most pronounced in Clade A, which includes a number of scansorial species that are difficult to diagnose on the basis of external morphology.

### Ancestral states analyses

The dated species tree for ancestral states analysis (Figs. 4 and 5) was congruent with our densely-sampled, undated molecular phylogeny (Fig. 3). Character states for Clade C were not included in most ancestral state analyses due to phylogenetic uncertainty and the

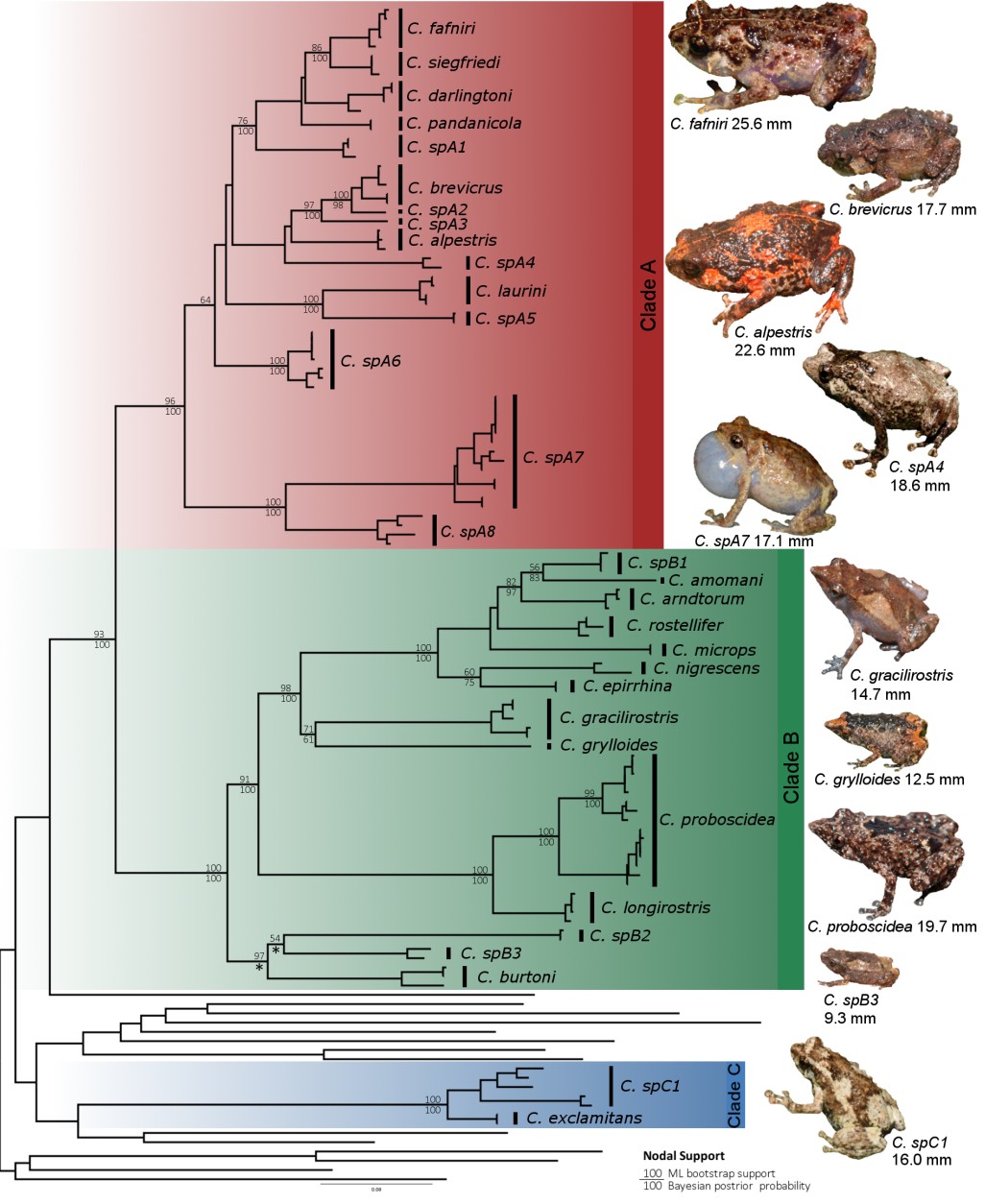

**Figure 3  Maximum likelihood tree of *Choerophryne*.** * indicates <50% Bayesian posterior probability, interspecific nodes without support values were poorly resolved in both analyses, intraspecific node supports are omitted for clarity. All photographs by S Richards.

relatively small number of lineages. These analyses highlighted the contrasting evolutionary trajectories of the two 'core' clades of *Choerophryne* (A & B). Clade A was inferred to have diversified primarily within montane habitats during the late Miocene (14 out of 15 nominal taxa), including more recent upslope shifts into mid and upper-montane zones (Fig. S2). Independent colonisation from eccentric origins to the North Papuan Mountains is inferred when geography is included (especially in the Foja Mountains) (Fig. 4A).

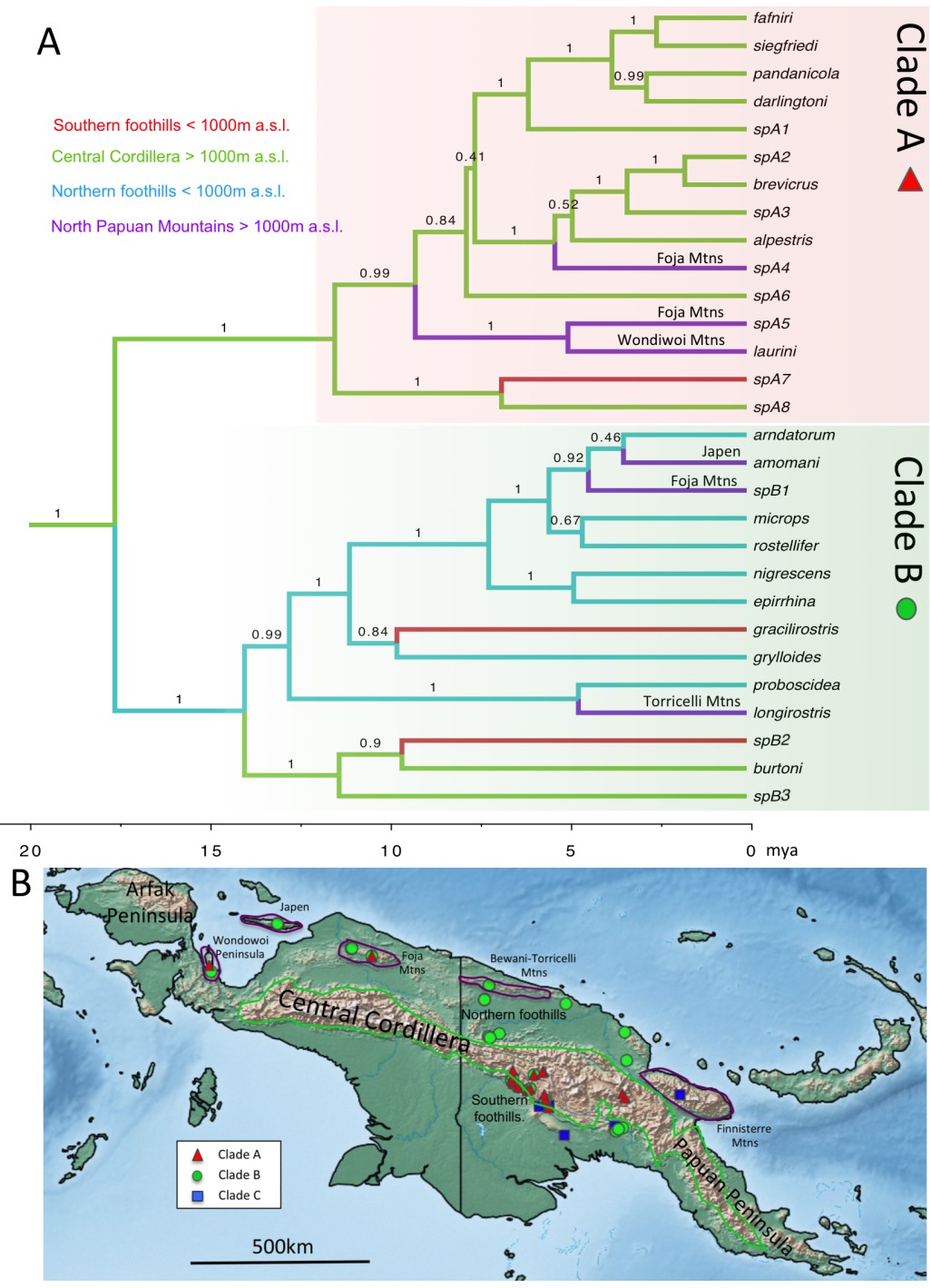

**Figure 4 Chronogram and distributional summary for _Choerophryne._** (A) Estimated using 12s and 16s data and rate-based calibration. Node values are Bayesian Posterior Support values from BEAST analysis. Axes along bottom indicate time in millions of years ago. Branches colour coded based on joint estimates of geographic region and elevation. Specific ranges in which inferred eccentric (Clade A) and centric (Clade B) endemics in the North Papuan Mountain ranges are annotated. (B) Map of main montane areas of New Guinea, and sampling localities for the three major clades of _Choerophryne._

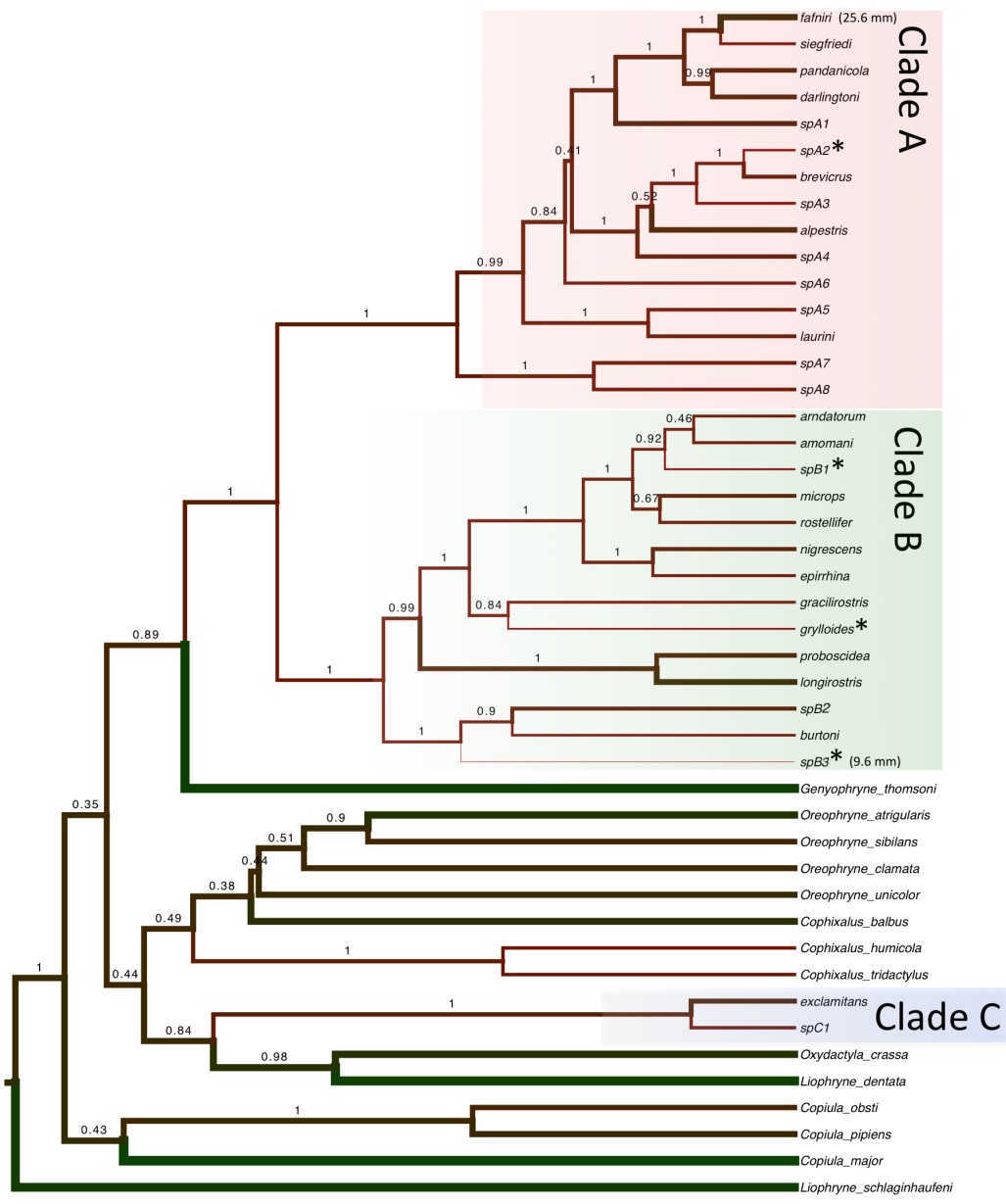

**Figure 5** **Body size evolution in *Choerophryne*.** Branch widths are proportional to maximum recorded adult male SVL. Green taxa are larger, red taxa are smaller. Miniaturised taxa (<15 mm) are indicated with an asterisk. Maximum recorded SUL of males in the genus *Choerophryne* ranges from 9.3 mm (spB3) up to 25.6 mm (*C. fafniri*).

In contrast Clade B was centred on hill forest habitats, but with 2–4 relatively recent upslope (centric) shifts into montane habitats in mostly distantly related taxa, again mainly occurring in isolated North Papuan Mountains (specifically Japen Island, and the Foja and Torricelli Mountains) (Fig. 4B, Fig. S2).

Miniaturised species (<15 mm) occurred across the phylogeny (Fig. 5), implying that multiple lineages of *Choerophryne* have independently evolved very small body size. Taxa in

the predominantly scansorial Clade A tended to be larger than those in the more terrestrial clade B.

Calling ecology was relatively labile across the genus, with multiple shifts between terrestrial and scansorial calling, the latter being inferred as the ancestral state for the common ancestor of clades A and B (Fig. S2 ). However, there were contrasting patterns across the two clades. Clade A was inferred as largely scansorial with a small number of shifts towards terrestrial calling, Clade B included a majority of taxa (9 out of 14) that call from on, or close to, the ground (this state was accordingly inferred as ancestral) with 3 transitions to scansorial calling.

### Phylogenetic regressions

All BayesTraits runs converged well before the burnin, and the concatenated runs yielded ESS of all parameters >1,000. In the analysis relating body size to ecology and/or elevation, both ecology and elevation (considered together following *Pagel & Meade, 2013*) exhibited significant phylogenetic structure (Lambda for all taxa was significantly positive: mean 0.55; 95% HPD = 0.12, 0.98). Ecology (as indexed by calling height) was positively associated with body size in the all taxa analysis (mean = 0.09, 95% HPD = 0.03, 0.15). In analyses focusing on specific clades this relationship was also positive, although the HPD included zero for Clade A (mean = 0.1; 95% HPD = −0.01, 0.19), but not Clade B (mean = 0.1; 95% HPD = 0.01, 0.20).

Elevation was not strongly related to body size in all relevant analyses, with a regression coefficient centred almost exactly on 0 when all taxa were included (mean = 0.01, 95% HPD = −0.11, +0.11). Analyses of the two main clades showed positive and negative relationships, although in both cases the HPD again included 0, suggesting the relationships were weak or insignificant (Clade A mean = 0.24, 95% HPD = −0.06, 0.5; Clade B mean = −0.05, 95% HPD = −0.20, 0.08).

Calling height was weakly negatively related to elevation, although in all cases the HPD again spanned 0 (all taxa mean = −0.65, 95% HPD = −1.27, 0.04; Clade A mean = −1.1, 95% HPD = −2.49, 0.15; Clade B mean = −0.74, 95% HPD = −1.62, −0.03). Removal of three high elevation (>2,500 m a.s.l.) taxa in Clade A that live in mossy grasslands where there are few arboreal habitats weakened this relationship further, resulting in a 95% highest probability posterior distribution that more broadly included 0 (mean = −0.47, 95% HPD = −1.06, 0.16).

## DISCUSSION

Despite the biological wealth and high endemism of the New Guinea Mountains (*Tallowin et al., 2017*) and emerging evidence for major evolutionary radiations (*Toussaint et al., 2014*; *Givnish et al., 2015*), only a small number of phylogenetic studies of lineages with distributions centred on the montane regions of New Guinea have been published (*Meredith et al., 2010*; *Toussaint et al., 2013*; *Irestedt et al., 2015*). Our study complements the recent work focusing on volant or large-bodied taxa, by presenting data for a lineage of small, direct-developing frogs that may be presumed to have comparatively low vagility.

## Species diversity and phylogeny

Molecular assessments of amphibian diversity on tropical islands over the last decade have revealed exceptionally high levels of previously unrecognised diversity (*Meegaskumbura et al., 2002*; *Vieites et al., 2009*). However, while New Guinea already has the most diverse insular frog fauna in the world (over 450 recognised species; *Frost, 2014*), molecular assessments of frog diversity in this region are scarce. While taxonomy was not the focus of this study, we uncovered 12 candidate species (although it remains possible that further work will show some of these to be conspecific with recognised taxa), in addition to three new taxa recently named (*Iannella, Richards & Oliver, 2014*; *Iannella, Oliver & Richards, 2015*). Molecular studies of other New Guinea microhylid frogs (*Mantophryne*) have also revealed a diversity of deeply divergent lineages (*Oliver et al., 2013*), and further fieldwork and integration of molecular, morphological and acoustic analyses seem certain to cement New Guinea's position as a global hotspot of amphibian diversity.

Clades A and B together formed a strongly supported monophyletic group, but the overall monophyly of all three sampled lineages of *Choerophryne* was not strongly supported (or rejected). There are however morphological synapomorphies uniting all three lineages of *Choerophryne* (see *Burton & Zweifel, 1995*), and their monophyly was also recently supported based on a phylogenomic study including examplars of all three major lineages (*Peloso et al., 2015*). The non-monophyly of *Choerophryne* in our analyses could be an artefact of rapid diversification and/or the short rapidly saturating loci used in this study. Resolution and further discussion of the phylogeny and generic taxonomy of *Choerophryne* will require larger nuclear gene-based datasets and sampling of taxa from other regions of New Guinea. However because of uncertainty in basal relationships, in this study we focused ancestral state analyses on the well-sampled and supported clades A and B.

There were distributional gaps in our genetic sampling (Fig. 4B). However, recent surveys in western New Guinea (upper Mamberamo, Fakfak mountains) have indicated that *Choerophryne* (which are usually easy to locate) are absent or rare, suggesting this disjunction reflects genuine absence (*Günther, 2000*; *Richards, Iskandar & Allison, 2000*). Another gap is the Papuan Peninsula, where endemic *Choerophryne* are found (Fig. S3). However, none of these taxa are shared with central New Guinea, suggesting that taxa in this region—which is geologically very distinctive—will have their own history. Furthermore, while future addition of taxa from this region into phylogenetic datasets is a research priority, we consider it unlikely to change the broadly reciprocal patterns of elevational distribution and montane colonisation between clades A and B in central New Guinea that we discuss below.

## Complex origins of montane endemism

Uplifting tropical mountains have been shown to be 'cradles' of young diversity in diverse regional bird communities (*Weir, 2006*; *Price et al., 2014*). Recent work on beetles, mammals and birds has suggested a similar association between diversification and the recent uplift of mountains in New Guinea (*Meredith et al., 2010*; *Toussaint et al., 2014*; *Irestedt et al., 2015*). In this study we complement such work by providing the first molecular phylogeny of a vertebrate clade (Clade A within *Choerophryne*) that is both

moderately diverse (15 nominal taxa), and almost entirely endemic to the New Guinea Highlands (>1,000 m a.s.l.). Our phylogeny suggests Clade A colonised lower montane habitats first, possibly by the mid-Miocene, while higher altitude taxa (i.e., >2,000 m a.s.l.) in Clade A are younger (Pliocene origin). This pattern is broadly consistent with progressive upslope colonisation as the Central Cordillera gained height through the late Miocene, and during the Pliocene, suggesting that recent mountain uplift has played a key role in the diversification of this lineage.

On the other hand we find weak evidence that New Guinea mountains have functioned as a 'museum'. One potential example from *Choerophryne* is a clade in the Central Cordillera region (*burtoni*, *sp*B2 and *sp*B3) that shows outwardly disjunct distributions and deep divergences (estimated 10 mya in our analyses). However, when compared to deeply divergent relict bird lineages or high phylogenetic endemism of mammals (*Jønsson et al., 2010*; *Rosauer & Jetz, 2015*) in the New Guinea mountains, our data do not at this stage provide strong evidence that relict taxa have inflated montane diversity in *Choerophryne*.

A further striking result of this study is the inference of both centric and eccentric origins of montane diversity in the younger, lower elevation, more isolated and poorly known North Papuan Ranges. These ranges are home to numerous endemic taxa or isolated populations (*Richards et al., 2009*; *Oliver et al., 2011*; *Oliver, Richards & Sistrom, 2012*; *Oliver, Richards & Tjaturadi, 2012*; *Oliver et al., 2016*; *Beehler et al., 2012*), but in most cases these are clearly related to, or even conspecific with, montane taxa occurring elsewhere in New Guinea (e.g., 100% of birds are allopatric isolates of lineages occurring in montane habitats elsewhere; *Beehler et al., 2012*). In *Choerophryne* two lineages in Clade A show a similar pattern; they appear to be endemic to montane habitats in the north Papuan Mountains (not found below around 1,000 m a.s.l.), related to taxa otherwise known only from montane Central Cordillera habitats, and unknown from the intervening lowlands (*Richards & Suryadi, 2003*). This apparent pattern of eccentric origins suggest that lower montane forests in New Guinea have a dynamic climatic history, including periods of major elevational depression similar to those inferred elsewhere in the tropics (*Colinvaux et al., 1996*; *Zhuo, 1999*).

However, ancestral state analyses of our well-sampled Clade B, also provides strong evidence for at least two and potentially three independent derivations of North Papuan montane endemics from surrounding lowland taxa (centric endemism) (Fig. 4A). Detailed fine scale sampling is required to understand the processes that have shaped this endemism; elevational segregation may be an outcome rather than a driver of speciation (*Caro et al., 2013*; *Freeman, 2015*). However, regardless of the exact process, this represents the first strong evidence that endemic montane vertebrates have arisen *de novo* in northern New Guinea from largely lowland lineages. These contrasting origins of endemism suggest that the young and isolated North Papuan Mountains may provide excellent opportunities for comparative analyses of the processes driving montane endemism in young tropical mountains.

Finally, mountain uplift may also inflate regional diversity at lower elevations by isolating formerly continuous populations of lowland taxa (vicariance). In New Guinea there is already compelling evidence that the uplift of the Central Cordillera has isolated northern

and southern vicars in lowland and aquatic taxa (*Rawlings & Donnellan, 2003*; *Georges et al., 2014*), and potentially also lower montane taxa (*Irestedt et al., 2015*). However, our sampling of *Choerophryne* did not reveal extensive north-south vicariance, although one possible exception is a recently described pair of potential sister taxa in Clade B from hill and lower montane forest; *C. gracilirostris* (south) and *C. grylloides* (north) that are estimated to have diverged around 10 mya. This general lack of signal for north-south vicariance is unsurprising given the majority of species in the two clades are associated with hill and montane forest, and are less likely to be isolated by mountain uplift than lowland or aquatic taxa.

## At the lower size limits of vertebrates; correlates of repeated miniaturisation

A number of anuran lineages that approach the minimum size limits for vertebrates have been recently described  (*Wollenberg et al., 2008*; *Kraus, 2010a*; *Rittmeyer et al., 2012*; *Lehr & Coloma, 2008*; *Kraus, 2011*; *Wollenberg et al., 2008*; *Rittmeyer et al., 2012*), and it has been suggested that miniaturised frogs may represent an often overlooked, but important ecological guild in tropical areas (*Rittmeyer et al., 2012*). Broadly, three patterns are globally apparent in miniaturised frogs: most lack a free-swimming tadpole stage (*Estrada & Hedges, 1996*); occur in wet tropical and usually insular regions; and are more-or-less terrestrial (*Kraus, 2010a*; *Rittmeyer et al., 2012*). Across the six different genera of Papuan microhylids that contain miniaturised taxa (*Aphantophryne*, *Austrochaperina*, *Choerophryne*, *Cophixalus*, *Oreophryne* and *Paedophryne*) all three of these correlates are evident.

Our analyses further indicate that within *Choerophryne* there have been at least three relatively recent shifts towards extremely small body size (three lineages with SUL ~15 mm or less), all of which are inferred in lineages that call on or close to the ground. This plasticity of body size and ecology of *Choerophryne* contrasts with conservatism of these same features in another miniaturised genus of Papuan microhylids, *Paedophryne* (*Rittmeyer et al., 2012*). Patterns of evolution across both genera do however strongly support the hypothesis that physiological or ecological constraints limit miniaturised taxa to a terrestrial lifestyle. Most recognised taxa missing from our analyses are moderate sized (SUL >15 mm) and scansorial, and likely belong in Clades A and C. Their inclusion in our analyses is unlikely to change the correlation between terrestriality and small size.

Counter to our initial prediction, we did not find a strong positive correlation between elevation and either ecology (calling height) or body size, as might be expected if desiccation risk is decreased at higher elevations (*Scheffers et al., 2013*). This lack of pattern may indicate that for frogs of extremely small size, physiological or ecological pressures associated with microhabitat use are a greater constraint on body sizes than elevation-related variation in climates. Unlike the correlation between terrestriality and small size in which we are confident, and which mirrors a broader pattern, further analysis including both *Choerophryne* taxa missing from our dataset, and other microhylid genera is probably needed to refine understanding of the potentially much more nuanced relationships between body size, ecology and elevation.

Finally, *Choerophryne* provides a striking example of an insular frog lineage that has undergone ecological diversification, with repeated shifts between scansorial and relatively terrestrial ecologies, reflected in significant reduction or even loss of terminal discs and shortening of limbs (*Günther, 2008*; *Kraus, 2010b*; *Günther & Richards, 2011*). Similar ecological diversity and morphological plasticity has also been observed in other microhylid lineages in New Guinea, as well as in other island systems such as Madagascar and the Philippines (*Andreone et al., 2005*; *Köhler & Günther, 2008*; *Blackburn et al., 2013*). In contrast, microhylids generally seem to be peripheral (and usually terrestrial or fossorial) components of frog diversity in continental regions (see *Duellman, 1999*).

## CONCLUSIONS

Our new phylogeny and ecophenotypic data for the microhylid frog genus *Choerophryne* indicates that montane areas have been colonised via a complex suite of biogeographic processes, especially upslope colonisation and speciation in presumably novel highland habitats and dispersal between montane islands, and that the relative importance of these processes has differed across even closely related lineages. *Choerophryne* also shows a correlation between extremely small size and utilisation of terrestrial habitats, mirroring a global pattern that suggests that, in frogs, ecological or physiological constraints largely limit extremely miniaturised taxa to terrestrial microhabitats in tropical areas.

## ACKNOWLEDGEMENTS

We thank the numerous landholders in New Guinea for access to their land, the National Research Institute and Department of Environment and Conservation (now Conservation and Environment Protection Authority) in Papua New Guinea for research and export approvals, the South Australian Museum for access to material in their care, and to the numerous other research organisations and NGOs that facilitated the collection and examination of material used in this study. We thank Mark Scherz and an anonymous reviewer for their extensive helpful comments on earlier versions of this manuscript.

### Funding

This work was supported by grants from the Australian Research Council to Paul Oliver, a McKenzie Postdoctoral fellowship to Paul Oliver from Melbourne University, and grant from the Australia Pacific Science Foundation to Paul Oliver, Mike Lee and Stephen Richards. The funders had no role in study design, data collection and analysis, decision to publish, or preparation of the manuscript.

### Grant Disclosures

The following grant information was disclosed by the authors:
Australian Research Council.
McKenzie Postdoctoral fellowship.
Australia Pacific Science Foundation.

## Competing Interests

The authors declare there are no competing interests.

## Author Contributions

- Paul M. Oliver conceived and designed the experiments, performed the experiments, analyzed the data, contributed reagents/materials/analysis tools, wrote the paper, prepared figures and/or tables, reviewed drafts of the paper.
- Amy Iannella conceived and designed the experiments, performed the experiments, analyzed the data, wrote the paper, prepared figures and/or tables, reviewed drafts of the paper.
- Stephen J. Richards conceived and designed the experiments, contributed reagents/materials/analysis tools, wrote the paper, reviewed drafts of the paper, collecting all material and field data.
- Michael S.Y. Lee conceived and designed the experiments, performed the experiments, analyzed the data, contributed reagents/materials/analysis tools, wrote the paper, reviewed drafts of the paper.

## Animal Ethics

The following information was supplied relating to ethical approvals (i.e., approving body and any reference numbers):

As this work involved museum preserved material, ethics approval was not sought.

## DNA Deposition

The following information was supplied regarding the deposition of DNA sequences:

GenBank accession numbers for all samples included in analyses are provided in Table S1.

## Data Availability

The raw data has been supplied as Supplementary Files.

## Supplemental Information

Supplemental information for this article can be found online at http://dx.doi.org/10.7717/peerj.3077#supplemental-information.

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
