# Peer review of "Mountain colonisation, miniaturisation and ecological evolution in a radiation of direct-developing New Guinea Frogs (Choerophryne, Microhylidae)"

_PeerJ, doi:10.7717/peerj.3077_

## Round 0.1 · original submission · Major Revisions

As you will see, both reviewers provided numerous thoughtful comments and suggestions throughout your text. Although there's much work to be done, I'm very optimistic that you'll be able to properly address all of the raised issues.

·

Basic reporting

See the general comments to the author.

Experimental design

See the general comments to the author.

Validity of the findings

See the general comments to the author.

Additional comments

Dear Editor,

The paper entitled ‘Mountain colonisation, ecological evolution and miniaturisation in a radiation of direct developing New Guinea Frogs (Choerophryne, Microhylidae)’ by Oliver et al. handles the evolution and ecology of Choerophryne frogs on New Guinea based on phylogenies produced based on two mitochondrial genes, and a series of morphological measurements as well as new and mined ecological and distributional data. On the whole I find the article to be a significant contribution to knowledge of a diverse and interesting radiation of microhylid frogs, providing another case study for the investigation of miniaturization in amphibians. The structure of the paper is logical, and the results are easy to understand. In my opinion the discussion is much too long, due to how thorough the authors were in discussing their results in the context of their predictions and the published literature. Most importantly, however, I have several comments and concerns regarding the contents of the article, which I detail below. Nevertheless, after major revisions, I recommend this article be accepted.
First a few minor complaints:
(1) The authors consistently misuse dashes, placing them most frequently where a comma should go. Em dashes (—) should be used much as you would use a semicolon; en dashes (–) to connect ranges, and hyphens (-) to bind words. Em dashes are usually not bordered by spaces. I have corrected this throughout, as this is a common error, seldom explained in full and even less frequently corrected.
(2) The authors adopted some incorrect emendations from Peloso et al. (see my comments on taxonomy below); these need to be fixed in the figures especially. Choerophryne is a feminine genus, whereas Albericus is masculine; names must be emended accordingly. Particular care needs to be paid to the figures and the supplemental files in making these changes.
(3) Measuring SVL (SUL) must either be done to the vent or to the urostyle tip consistently; the authors appear to have used a mixture of the two, but since they are not equivalent, mixing them will mislead analyses. Measurements should be carefully and specifically defined such that they can be repeated by others. The vent is generally a more reliable placemark, as the urostyle of some species is exceptionally short or long, and often cannot be seen when preserved frogs are pronate.
(4) Repeating BEAST runs to see if they agree with one another is not really an appropriate test of stationarity. Stationarity needs to be checked by examining the tree log in Tracer; cyclical jumps in optima, as sometimes occurs when there are multiple optima in parameter space, may not be revealed by simply repeating runs, but are immediately obvious in Tracer.
(5) Latin names were not italicised in the references section, presumably because the citations were generated with an automated system. Again, I have corrected this, but I wanted to bring this issue to the authors’ attention for future reference.
These errors are minor, and should require little work to change (or I have already changed them). A few more have been included in my comments on the manuscript file, which I attach. However, I have a few major concerns as well:
Firstly, the authors discuss at some length the distribution of their two clades A and B (see my comments on the taxonomy below), and even make inferences about where and when they differentiated and diversified, without performing an actual reconstruction of the phylogeography of the two groups. While it is true that the molecular data are not really adequate to make highly reliable phylogeographic inferences, they are still more poorly suited to dating using an arbitrary evolutionary rate. Without a decent calibration, such inferences are at best speculative, whereas a reconstruction of the phylogeography of the group would lend more substance to the ancestral state analyses of altitudinal distribution, which, without consideration of the phylogeography of the group, are more or less meaningless (consider: two sister species on separate mountains, where one occurs at higher altitude than the other. Without phylogeography, this is reconstructed as an upslope diversification, whereas with phylogeography, it is clear that they represent disparate evolutionary trajectories in different, allopatric environments). I recommend that the authors replace their discussion of the colonization history of the various mountains with respect to their ages, uplift, and climate oscillations, with an investigation of the phylogeography of the groups. The two are very similar, but the latter is more realistic, given the data available.
Secondly, through most of the manuscript, the authors contrast clades A and B, but include them in the same analyses. My concern is that signal is lost in their analyses because they are trying to examine one largely arboreal, generally larger lineage with very few instances of terrestriality, and one largely terrestrial lineage with fewer instances of arboreality, but more importantly several instances of miniaturization. In the section ‘Phylogenetic Regressions’, the authors are trying to derive overarching patterns across a relatively balanced (in terms of taxon number), ecologically disparate phylogeny. In consequence, the patterns that each of the two major clades are undergoing are obscured, resulting in mostly insignificant regressions. I strongly recommend that the authors re-run the analyses with their major clades independently. Below, I have outlined some important taxonomic information that the authors should consider here, which would give stronger justification to the analysis of clade A and clade B separately (though the A+B analysis should of course be kept as well, to give contrast).
Thirdly, the discussion: I will break this down section by section, to be thorough.
• It begins with a paragraph that re-states the context and essentially says ‘in the discussion, we are going to discuss things’. This is unnecessary, and could/should be deleted or condensed.
• The first section, ‘Unrecognised species diversity and phylogeny’, while informative, is not relevant to the actual aims of the study. It should be drastically shortened to focus on the fact (i) that there are several undescribed lineages worthy of recognition as candidate taxa, (ii) sequence-based studies of New Guinea’s amphibian fauna are really lacking, and (iii) clades A and B represent ‘Albericus’ and ‘Choerophryne’ in the old sense, with perhaps a brief discussion of their taxonomic status, and that of clade C (see my taxonomic comments below), with (iv) a short sentence stating that clade stability was such that the rest of the conclusions in the discussion are robust.
• ‘Geographic and taxonomic considerations’ could also be condensed and/or merged with the previous section. Fig. 5 is informative, but somewhat decreases the reader’s faith in the analysis, as it makes the quite decent sampling of the phylogenies look sparse by comparison. That being said, I think it is valuable and should be kept.
• The sections ‘Montane cradle or museum’, ‘Contrasting origins of montane endemics…’, and ‘Mountain uplift and vicariance’ are largely based on the questionable dates inferred in the tree. They do place the research in a good context, which is valuable, but if the authors agree to transition their focus to phylogeography rather than questionable dating, then these sections will need quite a lot of re-writing. I would also recommend the authors attempt to compress these three sections into one, dealing with the phylogeography, maybe including a little discussion of timing of events, and stating which hypotheses are supported, and which not supported, by the data. This will stand as a good counterpoint to the following section, focussed on correlates of repeated miniaturization.
• ‘At the lower size limits of vertebrates…’ is good. The authors need to check their definition of ‘extremely small body size’ against the literature; 15 mm is not particularly extreme—12 mm is much more so. This is of course a continuum however, and the authors are free to call these species extremely miniaturised if they please. They do need to bear in mind however that there are now quite a few frog species that have adult SVLs of less than 10 mm. See also my comments in the text, particularly in relation to the ‘specialised feeding apparatus’. The question of why microhylids are such good island colonisers is a particularly interesting one, for which I have not yet seen any convincing hypotheses.
• The conclusions are good, but probably need to be updated once the above has been taken into consideration.
Finally, I have a few remarks on taxonomic considerations that the authors should take under advisement: Choeroprhyne was recently rearranged by Peloso et al. in a paper that has already received some strong criticism for making unwarranted (and misinformed) taxonomic changes (see Scherz et al. 2016 Mol Phy Evol 100:372). The authors of the present article opted to focus on the evolutionary, rather than the taxonomic face of their research, which was certainly the right decision, given their aims. Nonetheless, I feel that neglecting to discuss the taxonomic implications of their new phylogeny is a significant shortcoming of the article. Peloso et al. united Albericus and Choerophryne based on sequences from four samples, two of which had NGS data available. These two were reconstructed by Oliver et al. to represent clades B (Choerophryne s.s.) and C (A. exclamitans), which are not closely related (albeit with low support). In Peloso’s study, the further two representatives of Albericus, from which no genomic data were available, clustered closer to Oliver’s clade B (Choerophryne s.s.) than to the basal clade C representative. In summary, Peloso et al. produced roughly the same tree topology as the present study, but they decided to homogenise the genus to one (Choerophryne), rather than split Albericus into two. Albericus exclamitans differs from all other Albericus species in its call type and at least one reliable morphological character (hidden tympanum in males). Albericus s.s. (to the exclusion of A. exclamitans) and Choerophryne are reconstructed as being reciprocally monophyletic by Oliver et al. and by Peloso et al., are morphologically diagnosable by a strong character, and are shown by Oliver et al. to also be ecologically distinct. Under this scenario, it seems that a more parsimonious option would be to keep Albericus and Choerophryne, and designate A. exclamitans as a new genus. At the very least, Albericus should be kept as a subgenus of Choerophryne, as it is reliably diagnosable. These options need to be carefully considered in the context of the taxon naming criteria proposed by Vences et al. 2013 Zootaxa 3636:201. While I concede that this taxonomic argument is only superficially relevant to the research at hand, it becomes important in the consideration of separate analyses of clades A and B, and in the conclusions derived therefrom. I thus recommend that the authors include a brief discussion of the taxonomic implications of their results either in their discussion, or in a supplementary information/appendix section.

I hope that the authors find this review helpful, and I wish them luck with their revisions. I hope to see more articles that take advantage of the fascinating evolutionary history of microhylids in the future.

Best regards,
Mark D. Scherz

Reviewer 2 ·

Basic reporting

See general comments.

Experimental design

Clearly identified and stated.

Validity of the findings

Clearly identified and stated.

Additional comments

I enjoyed reading this study. The authors have provided an interesting set of analyses on an understudied guild/taxa in an important hotspot for amphibians, paving the way for more taxonomic, ecological, and biogeographic research in this area. The results add further important data to broad-scale assessments of terrestrial anurans.

I have provided main comments in the returned text (Word doc.). Main points for consideration are:

• Some text requires rewording for clarity, and some statements require expansion/additional detail.
• There are some grammatical and punctuation errors. This needs to be checked, especially in the reference list.
• Was there evidence of intraspecific and/or interspecific variation in body size in the sampled frogs?
• I would like to see an explanation of how the work of Vieites et al (2009) regarding candidate species has been adopted.
• I would like presented the degree of sequence divergence for 16S, and how this relates within microhylids (and the Choerophryne), and to the adopted diagnoses of candidate species (re. previous point). The pairwise sequence divergence for all taxa used could be presented as SI.
• How was/was the sex of frogs used for morphometrics verified?
• This may not be appropriate/possible but calibrations of geographic events (e.g. uplift) could be used to investigate the validity of divergence estimates.
• The SI should include the MrBayes tree

Annotated reviews are not available for download in order to protect the identity of reviewers who chose to remain anonymous.

---

## Round 0.2 · Minor Revisions

As you will see, both reviewers were very satisfied with the new version of the manuscript, but still identified a number of small issues that need to be addressed before it can be accepted. But I believe you're very close.

·

Basic reporting

See general comments to the authors

Experimental design

See general comments to the authors

Validity of the findings

See general comments to the authors

Additional comments

Dear authors,

I am pleased to see that many of my recommendations from the first round of review have been addressed. In this second round of review I have found the paper reads much better, especially in light of the considerably condensed discussion. The figures are also much easier to interpret now. However, I have found a few small issues that still give me cause for concern. These are all small, and I strongly recommend this paper be accepted after these minor revisions.
1. I have several concerns with the supplemental files: these are submitted as a single word document containing multiple supplemental tables and figures, which is counter to the typical format employed by PeerJ. These should be divided into individual files. Additionally, the format of the figure legends is not entirely compliant with PeerJ's standard of having a title and then a legend for the figures.
2. I am concerned with the miniaturization-elevation hypothesis of the introduction, which is that smaller body size will correlate with higher elevation due to increased humidity and less desiccation risk in higher cloud forests. This trend is to my knowledge not known from other miniaturized frogs; in the microhylids of Madagascar, for example, several highly miniaturized species are found between 100–300 m a.s.l. It comes therefore as no surprise to me that this hypothesis is rejected, and may not be worth including in the introduction. I have expanded the ending of the second to last paragraph of the introduction to give this hypothesis more clearly, separating the trend from the hypothesis.
3. You currently do not make any statements in the results about taxon sampling, and do not explain the basis on which you exclude the candidate species they report from currently available names not included in their sample. Under normal circumstances this would not be of concern, but Clade A has more unnamed candidates than nominal taxa. It would be pertinent to have a sentence stating that the missing taxa are ruled out for these candidates on morphological/genetic/whatever basis.
4. C. epirrhina is misspelt in fig. 2.

I hope that these changes can be made quickly, and I look forward to seeing this article published.

Best regards,
Mark D. Scherz

Reviewer 2 ·

Basic reporting

No further comments

Experimental design

no further comments

Validity of the findings

No further comments

Additional comments

The authors have taken on board the comments, edits, and suggestions following the initial review, and I am satisfied that these have now been addressed and implemented in the manuscript.

There are still some minor editorial (punctuation, etc.) issues which have been noted in the attached document, and some suggestions re figures.

An interesting study which I look forward to reading the final copy of, upon publication.

Annotated reviews are not available for download in order to protect the identity of reviewers who chose to remain anonymous.

---

## Round 0.3 · accepted · Accept

I am happy with the final modifications to the manuscript.